# Question Answering as Programming for Solving Time-Sensitive Questions

**Xinyu Zhu**◇*    **Cheng Yang**◇    **Bei Chen**♡    **Siheng Li**◇
**Jian-Guang Lou**♡    **Yujiu Yang**◇

◇Shenzhen International Graduate School, Tsinghua University    ♡Microsoft

zhuxy21@mails.tsinghua.edu.cn    yang.yujiu@sz.tsinghua.edu.cn

{bei.chen, jlou}@microsoft.com

## Abstract

Question answering plays a pivotal role in human daily life because it involves our acquisition of knowledge about the world. However, due to the dynamic and ever-changing nature of real-world facts, the answer can be completely different when the time constraint in the question changes. Recently, Large Language Models (LLMs) have shown remarkable intelligence in question answering, while our experiments reveal that the aforementioned problems still pose a significant challenge to existing LLMs. This can be attributed to the LLMs' inability to perform rigorous reasoning based on surface-level text semantics. To overcome this limitation, rather than requiring LLMs to directly answer the question, we propose a novel approach where we reframe the **Q**uestion **A**nswering task **a**s **P**rogramming (**QAaP**). Concretely, by leveraging modern LLMs' superior capability in understanding both natural language and programming language, we endeavor to harness LLMs to represent diversely expressed text as well-structured code and select the best matching answer from multiple candidates through programming. We evaluate our QAaP framework on several time-sensitive question answering datasets and achieve decent improvement, up to 14.5% over strong baselines. [1]

## 1 Introduction

In our pursuit of knowledge and understanding, we often rely on factual questions to uncover the truth about the world around us. However, a critical aspect that is often overlooked is the presence of underlying temporal constraints within these questions. Time, an important dimension in the physical world, emerges as a ubiquitous and formidable constraint that can significantly impact the accuracy and relevance of the answer. Consider, for instance, the seemingly straightforward question: "Who is

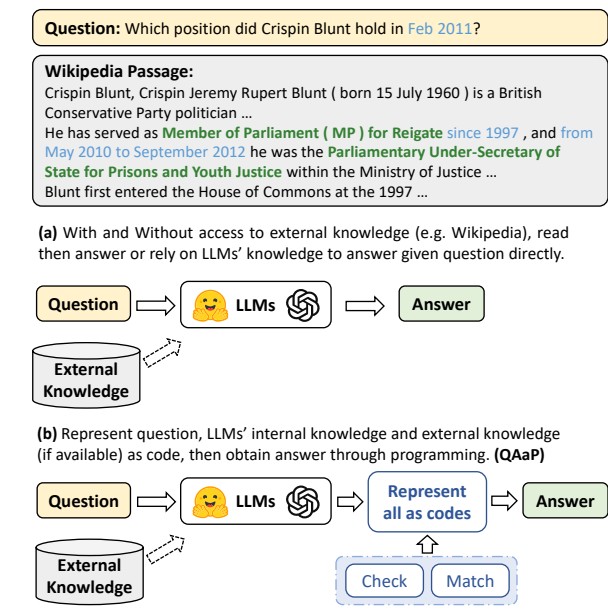

**Question:** Which position did Crispin Blunt hold in Feb 2011?

**Wikipedia Passage:**
Crispin Blunt, Crispin Jeremy Rupert Blunt ( born 15 July 1960 ) is a British Conservative Party politician …
He has served as **Member of Parliament ( MP ) for Reigate** since 1997 , and from May 2010 to September 2012 he was the **Parliamentary Under-Secretary of State for Prisons and Youth Justice** within the Ministry of Justice …
Blunt first entered the House of Commons at the 1997 …

**(a)** With and Without access to external knowledge (e.g. Wikipedia), read then answer or rely on LLMs' knowledge to answer given question directly.

**(b)** Represent question, LLMs' internal knowledge and external knowledge (if available) as code, then obtain answer through programming. **(QAaP)**

Figure 1: An example of a time-sensitive factual question from TimeQA (Chen et al., 2021): (a) illustrates the conventional process of question answering with LLMs, and (b) presents the proposed approach QAaP. The temporal information is colored with light blue and the potential answers are in **green**.

the current president of the United States?". While the answer may appear apparent, its validity depends entirely on the specific point in time when the question is posed. This realization underscores the importance of recognizing and accounting for these temporal constraints in our quest for reliable and meaningful information.

Recently, LLMs have exhibited excellent intelligence in question answering and challenged the dominance of traditional search engines. Nevertheless, despite their impressive performance, relying solely on LLMs for question answering still faces many shortcomings and limitations. As illustrated in Fig. 1 (a), current approaches to utilizing LLMs for question answering can be primarily categorized into two types: (1) relying on the internal knowledge of LLMs to generate direct answers,

---

* Work done during internship at Microsoft.

[1]Our codes and data are available at https://github.com/TianHongZXY/qaap

such as Chain-of-Thought (CoT) prompt (Wei et al., 2022) and (2) presenting LLMs with retrieved documents for reading and subsequent answering, such as ReAct (Yao et al., 2022). However, there are several challenges in answering time-sensitive factual questions with LLMs: 1) LLMs are insensitive to numbers (Nye et al., 2022) and struggle to comprehend the sequential, overlapping and inclusive relationships between dates, which can be attributed to their nature as probabilistic models and their inability to perform rigorous reasoning as symbolic systems; 2) it is difficult for LLMs to directly locate the relevant information and provide the correct answer, as relevant facts may often be scattered across a long document and expressed in diverse ways, as a result, simply searching for keywords in the question or relying on retrievers may fail to identify the relevant paragraphs; 3) it is very likely that the answer provided by LLMs fails to satisfy the constraint and it is hard to verify the correctness of the answer since they are given in natural language.

In this work, we endeavor to bridge the gap between existing approaches and the specific challenge posed by time-sensitive factual questions that require both rich world knowledge and intricate reasoning. To tackle the aforementioned challenges, we seek to harness LLMs' strong ability in natural language and programming language understanding to reframe the **Q**uestion **A**nswering task **a**s **P**rogramming (**QAaP**), depicted in Fig. 1 (b).

Specifically, our method consists of two phases: 1.Represent all as codes, since LLMs cannot rigorously reason by themselves, we endeavor to harness their strong coding ability to transform the question and context into well-structured codes. We first **Parse** the given question into a python dict, then **Extract** relevant information from the provided context and store these items in a python list. This allows a comprehensive gathering and organizing of relevant information dispersed throughout the documents. Besides, the collected code format information helps to facilitate the subsequent processing and avoid reasoning based on surface-level text semantics, 2.Choose answer through programming, due to the notorious hallucination inherent in LLMs, it is necessary to check if the extracted contents are faithful to the corresponding context. Furthermore, as there may be multiple potential answers, we need to reason out the best-matching answer to the question. Since all the obtained information is represented as codes, we can easily construct two functions **Check** and **Match** to reduce hallucination and ensure accuracy.

With our approach, we move beyond the traditional paradigm of using LLMs to directly generate an answer or read a given context and then provide an answer, avoiding the inability to verify whether the answer satisfies the constraints set out in the question. Furthermore, by storing intermediate information in code, we overcome the length limit of the model input, thus empowering LLMs to read through the passages and uncover the eligible answer concealed within the lengthy documents. Experimental evaluation on multiple time-sensitive question-answering datasets shows that our approach consistently outperforms strong baselines and approximates the performance of supervised methods. Notably, we achieve up to 14.5%, 10.5% and 8.6% absolute improvements on TimeQA, TempQuestions and TimeQuestions over state-of-the-art few-shot methods.

In summary, our contributions are as follows.

- Our experimental results reveal that LLMs cannot accurately answer factual questions with time constraints, even induced with a Chain-of-Thought prompt.
- We propose a novel reasoning approach for solving the aforementioned problem with LLMs, called **QAaP**, that reframes the question-answering task as programming by representing question and context as code, then obtaining the answer through programming, as opposed to asking LLMs to directly provide the answer.
- We demonstrate the superiority of QAaP compared to other few-shot methods on several datasets, which achieves up to 14.5% absolute improvement on TimeQA over strong baselines and surpasses the previous supervised SoTA on TempQuestions.

## 2 Related Work

### 2.1 LLMs augmented with tools

Recently, the rapid advancement of LLMs has brought great progress to the field of natural language processing (Brown et al., 2020; Hoffmann et al., 2022; Chowdhery et al., 2022). However, relying solely on LLMs' own capabilities is still limited, considering that human intelligence lies in the use of tools, many works have explored augmenting LLMs with tools (Schick et al., 2023; Mi-

alon et al., 2023). Cobbe et al. (2021) introduces an extra calculator to LLMs for effectively solving math word problems, and Gao et al. (2022a) applies retriever to obtain evidence for verifying the truthfulness of contents generated by LLMs. Similarly, Gou et al. (2023) employs various kinds of tools to correct the outputs of LLMs.

There have also been some works utilizing LLMs' coding ability to offload the solution to an external solver (Gao et al., 2022b; Chen et al., 2022; Lyu et al., 2023). However, there are several differences between those code-prompt works and ours: 1) prior works mostly were solving mathematical and symbolic problems, while in this work we mainly focus on answering the factual question that requires temporal reasoning; 2) prior works only utilized LLMs' own knowledge for problem solving, while we propose to apply LLMs' coding ability to represent both LLMs' internal knowledge and external knowledge (e.g., Wikipedia articles) as same-format codes, which enables us to easily enhance LLMs with different sources of knowledge, and also facilitates desired processing; 3) prior works did not explore verifying the correctness of LLMs-generated codes, while we propose to incorporate Check and Match steps to mitigate LLMs' hallucination and ensure accuracy.

## 2.2 Reasoning with LLMs

Reasoning ability is a hallmark of human intelligence, which allows adapting from limited data and accomplishing unseen sophisticated tasks. Nye et al. (2022) shows that letting language model output intermediate steps improves performance. Zhou et al. (2022) introduces Least-to-Most prompt that instructs LLMs to decompose a problem into multiple sub-problems and solve them one by one. Wei et al. (2022) proposes Chain-of-Thought (CoT) prompting, which induces LLMs to reason step by step and greatly boosts LLMs' reasoning ability. Following works try to improve CoT in many ways, including reducing human efforts in exemplar construction (Kojima et al., 2022; Zhang et al., 2022), improving faithfulness of reasoning process (Lyu et al., 2023), code style CoT (Gao et al., 2022b; Chen et al., 2022) and allowing LLMs to interact with outside environment (Yao et al., 2022).

All above works have showcased LLMs' remarkable capabilities in solving a wide range of complex reasoning tasks, including commonsense reasoning (Wei et al., 2022), mathematical reasoning (Zhu

et al., 2023) and symbolic reasoning (Zhou et al., 2022) tasks. In this work, we mainly solve time-sensitive factual questions, which are more like a cross of the former tasks. Although the content that the question asks may only relate to commonsense and world knowledge, it contains a strict symbolic constraint, which is time, and understanding temporal relationships requires advanced reasoning ability.

## 2.3 Temporal reasoning

There are many works that focus on temporal reasoning before the era of LLMs. Those methods are mainly based on KBQA systems (Talukdar et al., 2012; Jia et al., 2018b; Saxena et al., 2021) or MRC models (Zaheer et al., 2020; Izacard and Grave, 2021; Zhang and Yamana, 2022; Yang et al., 2022; Zhang et al., 2023b). There are many aspects in which our work differs from these earlier studies: 1) our method is few-shot, while they are mainly supervised; 2) we only use a simple Wikipedia search engine, while KBQA requires high-quality annotated knowledge bases; 3) we are able to verify if the constraint is satisfied, while MRC methods cannot. Nevertheless, these supervised methods are very strong and we include their results in the experiments.

## 3 Method

### 3.1 Task definition

In this work, we focus on solving time-sensitive factual questions with LLMs. The questions predominantly fall into four kinds of Wh-questions: 'What', 'Which', 'Where' and 'Who'. These types of questions seek specific information related to entities, events, and temporal aspects, forming the backbone of many real-world information-seeking scenarios.

As shown in the top left of Fig. 2, given a factual question $Q$ *"Salomón Rondón played for which team in Mar 2019?"*, which contains a time constraint $Q_t$ *"in Mar 2019"*, LLMs need to provide the appropriate answer $A$ *"Newcastle United"*, which best matches the constraint $Q_t$ set out in $Q$. We use $K_i$ and $K_e$ to refer to models' internal knowledge and external knowledge (e.g. Wikipedia) respectively. The context $C$ presented to a model can come from either $K_i$ or $K_e$ as shown in the left of Fig. 2. We use $EI$ to refer to the extracted information.

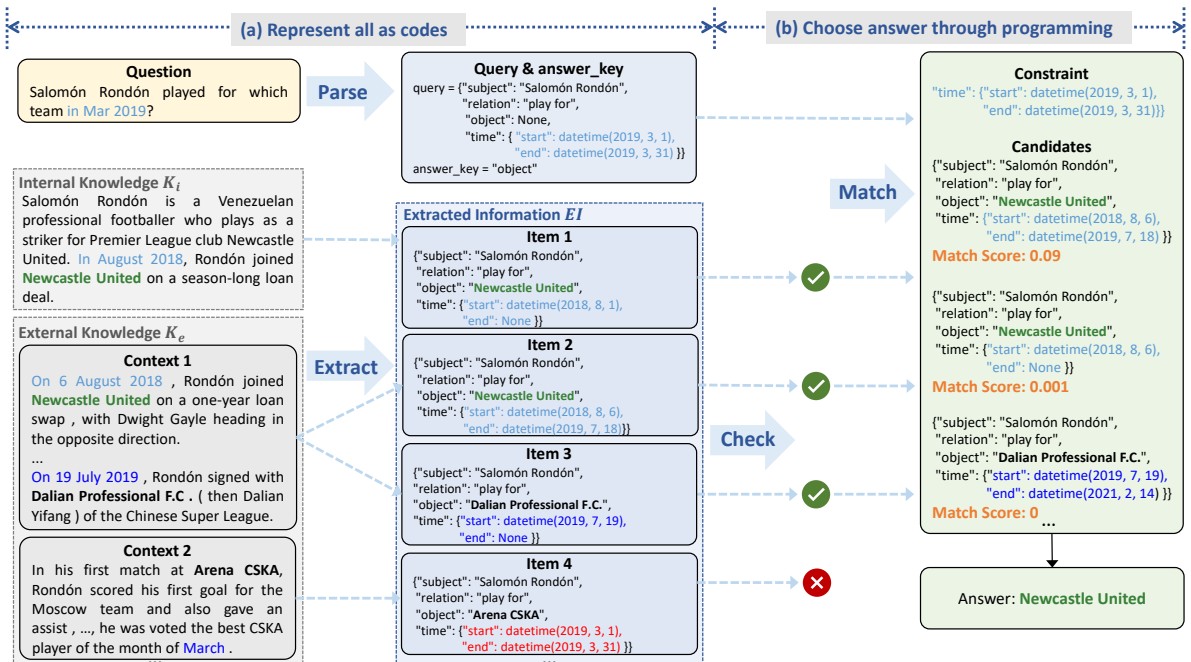

Figure 2: The whole framework of QAaP. Relevant and irrelevant temporal information is highlighted in light blue and blue, respectively. The correct answer is green; otherwise, it is red. Text related to potential answers is in **bold**.

## 3.2 Represent all as codes

**Parse.** We prompt LLMs to parse the given question $Q$ into a python dictionary variable named query $q$, which contains four keys: *subject*, *relation*, *object*, *time* and their corresponding value $s$, $r$, $o$ and $t$. In addition to the query, we define another variable answer_key to specify where the final answer should be placed at.

**Extract.** Given a text segment from the document either generated by LLMs or retrieved from Wikipedia as context $C_i$, we prompt LLMs to extract information related to the question as illustrated in the middle of Fig. 2. Each item of the extracted information $EI_i = \left\{ \left( s_j^i, r_j^i, o_j^i, t_j^i \right) \right\}_{j=1}^{N}$ is also represented as a python dictionary similar to the query and stored in a predefined python list information, so as to comprehensively gather relevant information that appears in different parts of a long document.

## 3.3 Choose answer through programming

As mentioned before, LLMs are not sensitive to numbers and often suffer from hallucinations. For example, LLMs may trickily fill in the time that appeared in the question into the extracted items regardless of the context, resulting in a perfectly temporal matching but incorrect answer. To cope with such a phenomenon, thanks to the fact that

the extracted information is presented in code form, we can easily construct two functions **Check** and **Match** to verify the faithfulness of extraction and find the best matching answer.

**Check.** We first check if the extracted item is in the same format as query, which ensures the answer is correctly placed at answer_key, and that the extracted time should appear in the corresponding context. When external knowledge $K_e$ is accessible, we also check whether the items extracted from LLMs' internal knowledge have shown up in those extracted from $K_e$ because LLMs often generate fake facts.

As exemplified in Fig. 2, the answer *Newcastle United* generated by LLMs also appears in the Wikipedia passage, therefore we keep the extracted item. Note the middle bottom of Fig. 2, the extracted time *2019* of the 4th item does not show up in the corresponding Context 2 and therefore it is removed during the Check step.

**Match.** In order to choose the best matching answer from numerous candidate answers, we employ the intersection over union (IoU) of time between the question $Q$ and the candidate $X$ as the match score, which is defined in Eq. (1), and sort the candidates according to it. When the question specifies both the start $Q_{t_s}$ and the end $Q_{t_e}$ of the time constraint, this measurement quantifies the degree of temporal alignment between the question and each

candidate. If the time constraint in query only contains a start or an end, we choose the inverse of the absolute value of the differences between $Q_{t_s}$ and $X_{t_s}$ or $Q_{t_e}$ and $X_{t_e}$ as match score.

$$\text{MS}(Q, X) = \frac{\min(Q_{t_e}, X_{t_e}) - \max(Q_{t_s}, X_{t_s})}{\max(Q_{t_e}, X_{t_e}) - \min(Q_{t_s}, X_{t_s})} \quad (1)$$

Finally, we determine the answer by selecting the candidate that achieves the highest score. The Match step ensures that the chosen answer aligns closely with the time constraint specified by the question, thus guaranteeing the accuracy and relevance of the answer. We also discuss applying LLMs for accomplishing the Check and Match functions in Sec. 4.3.4.

## 4 Experiments

### 4.1 Experimental setup

#### 4.1.1 Datasets

We consider several widely-used temporal reasoning datasets: TimeQA (Chen et al., 2021), TempQuestions (Jia et al., 2018a) and TimeQuestions (Jia et al., 2021).
**TimeQA** is a time-sensitive question answering dataset curated by extracting time-evolving facts from WikiData and aligning them with Wikipedia pages with the help of human workers. The dataset comprises two subsets with different levels of difficulty. The easy subset contains documents with asked temporal information explicitly stated, while the hard subset requires more advanced reasoning skills as the temporal information is implicit and cannot be retrieved through keyword searching.
**TempQuestions** and **TimeQuestions** are two similar temporal QA datasets compiled from a variety of general-purpose KG-QA benchmarks. Questions are divided into four classes: explicit temporal, implicit temporal, temporal answer and ordinal constraints. We choose the explicit subset and remove the questions without any temporal signal. This is because implicit questions require extra steps to uncover the implicit time indicated by the things/events, while we only use a simple Wikipedia search engine and there are so many ambiguous things that have the same name (e.g., song, movie, person name, ...), it is difficult for LLMs to determine which entity the question is referring to. Therefore, strong retrievers are needed to retrieve relevant documents. It is worth noting that our method is agnostic to the way of retrieving external documents and can be seamlessly combined with

those off-the-shelf retrievers, which we leave for future work.

#### 4.1.2 Baselines

For comparison under a few-shot setting, we employ the well-known method CoT (Wei et al., 2022) as the baseline. In addition, we include a recently popular method ReAct (Yao et al., 2022), which incorporates external knowledge and allows LLMs to execute search and lookup actions to interact with the Wikipedia website. Finally, we present results from state-of-the-art fine-tuning models for a comprehensive comparison.

#### 4.1.3 Implement details

To fully utilize LLMs' internal knowledge $K_i$, following Yu et al. (2023) we elicit $K_i$ by prompting LLMs to generate a background document for the given question. Additionally, in order to incorporate external knowledge $K_e$, we add a search process to the prompt and apply a Wikipedia search engine similar to ReAct. We treat the documents from $K_i$ and $K_e$ in the same way. For all experiments, we use gpt-3.5-turbo as the backbone model unless otherwise specified. Details of prompt can be found in Appendix C.

### 4.2 Main results

Table 1 presents the main results on the three temporal reasoning datasets. QAaP consistently demonstrates superior performance across all the datasets under a few-shot setting. While CoT performs well on the TempQuestions dataset by relying solely on the LLMs' internal knowledge, it falls short on TimeQA and TimeQuestions. This discrepancy can be attributed to the limited coverage of the LLMs' $K_i$ in memorizing the rich facts of the open world. On the other hand, despite having access to external knowledge $K_e$, ReAct still underperforms, indicating its potential shortcomings in locating relevant information from provided documents. Impressively, QAaP outperforms other methods on all the datasets, exhibiting substantial improvements. Specifically, it achieves a remarkable enhancement of $13.9\%$, $14.5\%$, $10.5\%$, and $8.6\%$ in exact match scores on TimeQA-Easy, TimeQA-Hard, TempQuestions, and TimeQuestions, respectively. The above experimental results clearly demonstrate the effectiveness of QAaP.

The improvement might be explained by two reasons: one is that with QAaP we are able to comprehensively extract relevant information from the

| Backbone | Method | TimeQA-Easy | | TimeQA-Hard | | TempQuestions | | TimeQuestions | |
|---|---|---|---|---|---|---|---|---|---|
| | | EM | F1 | EM | F1 | EM | F1 | EM | F1 |
| Fine-tune | | | | | | | | | |
| | Previous SoTA | 60.5 | 67.9 | 46.8 | 54.6 | 43.8 | 44.6 | 56.8 | - |
| Few-shot gpt-3.5-turbo | CoT | 24.6 | 34.2 | 21.7 | 30.4 | 49.8 | 60.1 | 28.2 | 41.2 |
| | ReAct | 34.3 | 41.5 | 25.1 | 29.4 | 28.0 | 33.7 | 12.6 | 17.5 |
| | QAaP | **48.2** | **58.3** | **39.6** | **49.3** | **60.3** | **68.1** | **36.8** | **46.7** |

Table 1: Results on temporal reasoning tasks. The previous SoTA supervised models for TimeQA, TempQuestions and TimeQuestions are FiD (Chen et al., 2021), TEQUILA (Jia et al., 2018b) and EXAQT (Jia et al., 2021) respectively. The best scores are in **bold** and the second are underlined.

| Method | $K_e$ | TimeQA | | TempQuestions |
|---|---|---|---|---|
| | | Easy | Hard | |
| CoT | ✗ | 24.6 | 21.7 | 49.8 |
| CoT-variant | ✔ | 22.0 | 18.5 | 48.8 |
| QAaP | ✔ | **48.2** | **39.6** | **60.3** |
| CoT-variant* | ✔ | 48.0 | 33.7 | - |
| QAaP* | ✔ | **52.6** | **46.7** | - |

Table 2: Ablation study of providing enough context in CoT. Results are exact match scores. $K_e$ refers to external knowledge. * indicates given golden paragraphs containing ground truth.

| Check | | TimeQA | | TempQuestions |
|---|---|---|---|---|
| $t$ | $K_i$ | Easy | Hard | |
| ✗ | ✗ | 32.8 | 27.6 | 55.2 |
| ✗ | ✔ | 45.0 | 37.0 | 59.3 |
| ✔ | ✗ | 39.8 | 32.1 | 56.6 |
| ✔ | ✔ | **48.2** | **39.6** | **60.3** |

Table 3: Ablation study of checking different aspects.

provided knowledge sources and store it efficiently, which facilitates the subsequent processing steps. The other is that rather than relying solely on LLMs to reason over raw context, QAaP chooses the final answer through programming. Since all the things are represented as codes, QAaP minimizes the need for human intervention to check the extracted contents and find the best matching answer.

However, there is still a non-negligible gap compared to the best-supervised models, which validates the difficulty of this task and indicates more efforts are needed in future work.

## 4.3 Ablation study

### 4.3.1 Can LLMs find the answer provided enough context?

To investigate if LLMs can answer factual questions with time constraints provided enough context, we implement a variant of few-shot CoT by feeding it with the same context used for our approach. Specifically, we present the text segments to them sequentially and ask if they can answer the question based on the context until they say yes and give the answer. Moreover, we lower the difficulty by using golden paragraphs as context, which consists of the paragraph containing the ground truth and three paragraphs before and after it.

As demonstrated in Table 2, though given the same context, QAaP achieves a much higher exact match score compared to directly reading and answering. The performance of the CoT-variant is even worse than CoT, which exposes LLMs' limitations in discriminating the answer that meets the constraint from the other irrelevant ones. Significant performance improvements are observed when golden paragraphs are provided. This result highlights the challenges faced by LLMs in accurately identifying the correct answer from lengthy documents. Notably, QAaP surpasses CoT-variant*, even though the latter directly accesses the golden paragraphs. This finding further underscores the advantage of our proposed method.

### 4.3.2 How much does Check alleviate hallucination?

To examine the effect of Check-in alleviating hallucination, we explore it along with two axes: 1) check if the time $t$ of extracted items appears in context and 2) check if the items extracted from LLMs' internal knowledge $K_i$ are also included in those extracted from external knowledge $K_e$.

As shown in Table 3, compared to checking the extracted time, checking the contents obtained from $K_i$ can bring more improvement. This discovery implies that the degree of hallucination is more severe when LLMs are allowed to generate content directly. However, even when conditioned on a given context, LLMs may still make up contents

| Match | Check | TimeQA | | TempQuestions |
|---|---|---|---|---|
| | | Easy | Hard | |
| ✗ | ✗ | 34.2 | 26.7 | 54.2 |
| ✗ | ✔ | 48.1 | 36.4 | 59.3 |

Table 4: Ablation study of letting LLMs choose the best answer from extracted candidates.

| Model | TimeQA | | TempQuestions |
|---|---|---|---|
| | Easy | Hard | |
| ChatGPT | 47.9 | 38.6 | 56.9 |
| GPT-4 | **48.2** | 38.5 | 56.9 |
| Manual | **48.2** | **39.6** | **60.3** |

Table 5: Employing LLMs for constructing Check and Match functions. ChatGPT and GPT-4 can be accessed via https://chat.openai.com.

that do not align with the provided context. The experimental results affirm the vital role played by the Check step in mitigating hallucination, shedding light on the substantial presence of hallucination in LLMs. These findings emphasize the importance of verifying the contents generated by LLMs to guarantee accuracy and reliability.

### 4.3.3 Can LLMs directly identify the best matching answer to the question?

We are curious about whether it is trivial for LLMs to find the best matching answer given the parsed query and extracted information. To this end, we remove the Match step and prompt LLMs to provide the answer through in-context learning.

The results are presented in Table 4, it can be observed that without the Match step, the exact match score relatively decreased by 8.1% on TimeQA-Hard, which stresses the indispensability of the Match step when the constraint necessitates rigorous reasoning to satisfy. In contrast, there is only a slight drop in performance on TimeQA-Easy and TempQuestions. This can be explained by the fact that we did not design a very sophisticated Match function for selecting the best answer. Additionally, the strong result of directly matching further substantiates the advantages of representing the question and context as codes, which can facilitate better reasoning for LLMs and significantly improve the accuracy compared to reading then answering methods.

It is worth noting that the Check step plays a key role in the final performance when the Match step is absent. Without the Check and Match steps, LLMs face challenges in directly identifying the

most appropriate answer from multiple candidates, as there are many interfering items. The results indicate that these two steps are complementary and indispensable.

### 4.3.4 Are LLMs capable of writing qualified Check and Match functions?

In our previous experiments, the Check and Match functions were manually constructed. However, in this section, we aim to explore the potential of leveraging LLMs to perform these tasks. Specifically, we provide clear instructions for the Check and Match processes, then we randomly select and concatenate an instance $\{Q, q, C, EI\}$ obtained from previous steps to the instruction as input prompt, finally we present the prompt to the LLMs and require them to write the desired functions. The results, as shown in Table 5 demonstrate the feasibility of replacing human effort with LLMs. Both ChatGPT and GPT-4 exhibit the capability to comprehend the task and provide an eligible solution.

This promising result suggests the potential direction for future research to explore how LLMs can self-verify their answers through programming. While LLMs have shown impressive performance in many NLP tasks, ensuring the accuracy and reliability of their generated responses remains a challenge. Exploring the integration of these verification mechanisms within LLMs can lead to more robust and trustworthy question-answering systems.

### 4.4 Case study

A typical exemplar is presented in Fig. 3. CoT encounters hallucinations and makes up facts that don't exist, which exposes its limitation when the question asked is beyond its scope of knowledge. Even given the golden paragraphs, CoT-variant* fails to reason correctly and mistakes Margrete Aamot Øverland as the final answer. ReAct tries to find the relevant sentences by looking up keywords, however, neither the year nor the keyword in the question appears in the passage. This is one of the challenges we mentioned earlier, without strong retrievers LLMs are usually unable to solve these questions, resulting in performance upper bounded by external retrievers.

As a comparison, results generated by QAaP are shown at the bottom. We omit the process of Check and Match steps for the simplicity of illustration. First, LLMs parse the question into a Python dictionary named query and generate a relevant background document with their internal knowledge.

QUESTION: Who was Arnulf Øverland 's spouse between May 1944 and Jun 1944?
CONTEXT: Arnulf Øverland was a Norwegian author, poet, and critic.
...
Personal life . In 1918 he had married the singer **Hildur Arntzen** ( 1888–1957 ) . Their marriage was dissolved in 1939 .
In 1940 , he married **Bartholine Eufemia Leganger** ( 1903–1995 ) . They separated shortly after , and were officially
divorced in 1945. Øverland was married to journalist **Margrete Aamot Øverland** ( 1913–1978 ) during June 1945.
GOLDEN ANSWER: **Bartholine Eufemia Leganger**

---

COT (WITHOUT CONTEXT):
Arnulf Øverland was a Norwegian author, poet, and social commentator. He was married to pianist and composer
**Harriet Cohen** from 1937 until her death in 1967. **There is no answer.** ✗

---

COT-VARIANT* :
Context: <CONTEXT>
Is the answer in the context? Yes
Answer: **Margrete Aamot Øverland** ✗

---

REACT:
Thought 1: I need to search Arnulf Øverland and find out who his spouse was between May 1944 and Jun 1944.
Action 1: Search[Arnulf Øverland]
Observation 1: Arnulf Øverland. Ole Peter Arnulf Øverland ( 27 April 1889 – 25 March 1968 ) was a Norwegian poet
and artist. He is principally known for his poetry which served to inspire the Norwegian resistance movement during
the German occupation of Norway during World War II .
Thought 2: The paragraph does not tell who Arnulf Øverland's spouse was between May 1944 and Jun 1944, maybe I
can look up the years between 1944 and 1944. First I can look up "1944".
Action 2: **Lookup[1944]**
Observation 2: No more results.
Thought 3: There is no results for "1944", let's look up "spouse".
Action 3: **Lookup[spouse]**
Observation 3: No more results.
Thought 4: The passage does not tell who Arnulf Øverland's spouse was between May 1944 and Jun 1944.
Action 4: **Finish[]** ✗

---

QAAP:
Question Parsing:
```python
query = {"subject": "Arnulf Øverland", "relation": "spouse", "object": None, "time": {"start":
datetime(1944, 5, 1), "end": datetime(1944, 6, 30)}}
answer_key = "object"
```
Generate a background document from Wikipedia to answer the given question: Arnulf Øverland was a Norwegian
author, poet, and critic. He was married to the author and translator **Berit Ås** from 1945 until his death in 1968.
Extract information relevant to the query:
```python
information.append({"subject": "Arnulf Øverland", "relation": "spouse", "object": "Berit Ås",
"time": {"start": datetime(1945, 1, 1), "end": datetime(1968, 12, 31)}})
```
Context: <CONTEXT>
Extract information relevant to the query:
```python
information.append({"subject": "Arnulf Øverland", "relation": "spouse", "object": "Hildur
Arntzen", "time": {"start": datetime(1918, 1, 1), "end": datetime(1939, 12, 31)}})
information.append({"subject": "Arnulf Øverland", "relation": "spouse", "object": "Bartholine
Eufemia Leganger", "time":{"start": datetime(1940, 1, 1), "end": datetime(1945, 12, 31)}})
information.append({"subject": "Arnulf Øverland", "relation": "spouse", "object": "Margrete
Aamot Øverland", "time": {"start": datetime(1945, 6, 1), "end": None}})
```
---
**<Check>**
**<Match>**
Best matching answer: **Bartholine Eufemia Leganger**. ✔

Figure 3: Examples of correct and incorrect answers were obtained with different methods on TimeQA. Interference
answers or wrong parts and time information not related to the question are highlighted in red and blue respectively.
Correct answer and relevant time are in green and light blue. The text implies potential answers is in **bold**.

Here again, the LLMs hallucinated a non-existing
fact that *"Arnulf Øverland was married to Berit
Ås"*. After extracting the relevant information, we
perform the Check step to verify the faithfulness of
extraction and the reliability of contents generated
by LLMs. As the extracted information is struc-
tured as code, we can easily filter the wrong and
fake items and therefore mitigate the hallucination,
which is proved in Sec. 4.3.2. Finally, we choose
the best matching answer through the Match step.

## 5 Discussion and Future Work

Our findings reveal the inherent difficulty faced by
LLMs in accurately answering seemingly straight-
forward factual questions when they involve spe-
cific time constraints. This can be attributed to the
nature of neural networks. A well-known prob-
lem with neural networks is that they are black-box
models and do not have symbolic reasoning ca-
pabilities. Consequently, even the most powerful
language models may struggle to perform rigor-
ous reasoning. However, in this work, instead of

relying solely on LLMs to directly provide the answer, we reframe the question-answering task as programming. By leveraging LLMs to represent the question and context as codes, we can easily alleviate the hallucination and improve answer accuracy through the subsequent Check and Match processes.

We hope our work can shed light on future work in exploring how to enhance LLMs' reasoning ability with more advanced means and tools. Additionally, we anticipate there will be more human-in-the-loop approaches to effectively reduce the hallucination while only requiring minimal labor efforts, for example, incorporating structured knowledge to automate the verification process. Moreover, we mainly focus on solving factual questions with time constraints, but many factual questions in real life may contain various types of constraints, such as number, order, location, etc. For different tasks, our method can be easily adapted to deal with other types of constraints, we just need to represent the constraint into an appropriate class in python and define the metric that measures how well is the constraint satisfied in the Match function. Furthermore, it is convenient to cope with different kinds of hallucination by incorporating additional check processes into the Check function. We leave that for future work.

We view our effort as the first step towards solving open-domain questions with some kinds of strict constraints and we hope this work can inspire other work to tackle questions with constraints in more general scenarios. A promising research direction is to enable LLMs to achieve comparable accuracy rates to search engines while maintaining the simplicity of interaction, which will greatly boost the practicality of LLMs.

## 6 Conclusion

In this work we propose a novel approach, **QAaP** (**Q**uestion **A**nswering **a**s **P**rogramming), to tackle the challenges posed by time-sensitive factual questions. By leveraging LLMs' exceptional abilities in natural language understanding and programming, QAaP can transform diversely expressed text into well-structured codes, enabling LLMs to capture both the desired knowledge and the underlying constraints, particularly the temporal aspects. Experiments demonstrate that existing LLMs face significant difficulty in effectively comprehending the temporal constraint stated in the question. While

our approach consistently demonstrates superior performance over strong baselines with LLMs. We hope this work can shed light on the future research direction on enhancing LLMs' reasoning ability to tackle real-world questions with various constraints and developing more efficient methods to reduce the hallucinations LLMs frequently encounter.

## Limitations

The results on multiple datasets demonstrate the effectiveness of our proposed framework QAaP, which can improve the accuracy of LLMs in answering time-sensitive factual questions. However, there are still limitations in our work:

1) We do not evaluate our method on other question-answering datasets with different kinds of constraints. The main reason is that there are limited relevant datasets. We also do not include question-answering datasets that require multi-hop retrieval to collect enough related documents like hotpotQA (Yang et al., 2018) since a strong retriever is needed and the accuracy may be mainly depended on the retriever, but our approach is agnostic to the way of introducing external knowledge and can be combined with off-the-shelf retriever seamlessly.

2) When solving questions requiring only commonsense or world knowledge, QAaP may not be necessary because there is no constraint in the question that needs rigorous reasoning to satisfy. This limitation can be also found in Table 1 of Lyu et al. (2023) where faithful-CoT does not help on StrategyQA dataset (Geva et al., 2021).

3) We only report experimental results with one backbone model `gpt-3.5-turbo`. This is mainly due to the high cost of other OpenAI models. However, to further prove the effectiveness of our method, we conduct some small-scale experiments with `text-davinci-003` and include the results in Appendix B, which verifies that our approach performs better than other baselines with different backbone models. We leave exploring other open-source models (Wang et al., 2022) for future work.

## Ethics Statement

In this work, our proposed approach QAaP can effectively solve factual questions requiring temporal reasoning, however, there is still a potential social risk. The main concerning risk of our work may be the hallucination of utilized LLMs (Zhou et al.,

2021; Ji et al., 2023; Zhang et al., 2023a), since the method we develop is aimed at answering factual questions, when it is applied to LLMs that are deployed into real production environments, it may provide erroneous and misleading answers to users due to the hallucination happened during the Parsing and Extract step. Similar challenges exist in the computer vision field too (He et al., 2023a,b). This can be tackled by designing more sophisticated Check and Match steps, which we also highlight in previous experiments and is proved effective in alleviating hallucination, thus helping reduce potential risks posed to society.

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

| Dataset | Split | # of samples |
|---|---|---|
| TimeQA | Easy | 2997 |
| TimeQA | Hard | 3078 |
| TempQuestions | Explicit | 297 |
| TimeQuestions | Explicit | 942 |

Table 6: Dataset statistics.

## A Dataset Statistics

Dataset statistics are shown in Table 6.

## B Additional Experiments

We conduct experiments with `text-davinci-003` and present the results in this section. Due to the high cost of OpenAI APIs, we only sample 100 questions for each dataset with random seed set to 0. As shown in Table 7, the results confirm the effectiveness of our method.

## C Full Prompts

TimeQA prompt is 3-shots and divided into three parts illustrated in Fig. 4, Fig. 5, Fig. 6. We have included other prompts in the supplementary materials and will open source all of them for further research.

| Backbone | Method | TimeQA-Easy | | TimeQA-Hard | | TempQuestions | | TimeQuestions | |
|---|---|---|---|---|---|---|---|---|---|
| | | EM | F1 | EM | F1 | EM | F1 | EM | F1 |
| Few-shot `text-davinci-003` | CoT | 17.0 | 26.2 | 15.0 | 28.0 | 39.0 | 49.6 | 33.0 | 44.8 |
| | ReAct | 30.0 | 42.7 | 26.0 | 37.4 | 33.0 | 42.2 | 22.0 | 31.6 |
| | QAaP | **38.0** | **50.1** | **38.0** | **49.4** | **47.0** | **51.9** | **40.0** | **48.9** |

Table 7: Results on temporal reasoning tasks with `text-davinci-003`. The best scores are in **bold** and the second are underlined.

---

Solve a question answering task by first parsing the question, figuring out what the question is asking, representing the query as a python dictionary with the key where the answer should be stored specified as either the subject or object, with the initial value of None. Secondly, decide what entities to search to find the answer. Thirdly, generate a background document that contains information relevant to the question being asked. Then read the generated document and the searched passage, step by step extract the information that directly and explicitly relates to the question, place the answer as the value of the answer key in the dictionary. For example, 'XXX joined A team in 2017, ..., in 2019, B team signed a contract with XXX', it is easy to know that B team and A team are mutually exclusive, therefore the termination time of A team is in 2019. Represent the extracted information as a dictionary and adding it to a list. If the context does not tell any useful information, extract nothing.
Here are some examples.

**Question:Which school did David Jolly go to in Jan 1989?**

Question parsing:
```python
query = {"subject": "David Jolly", "relation": "go to school", "object": None, "time":
{"start": datetime(1989, 1, 1), "end": datetime(1989, 1, 31)}}
answer_key = "object"
```

Search:
```python
entities_to_search = ["David Jolly"]
```

Generate a background document from Wikipedia to answer the given question: David Jolly is an American politician who served as the U.S. Representative for Florida's 13th congressional district from 2014 to 2017. He graduated from Indian Rocks Christian School in 1990.
Extract information relevant to the query:
```python
information.append({"subject": "David Jolly", "relation": "go to school", "object":
"Indian Rocks Christian School", "time": {"start": datetime(1986, 1, 1), "end":
datetime(1990, 12, 31)}})
```

Context: David Jolly David Wilson Jolly ( born October 31 , 1972 ) is an American attorney , former lobbyist , and politician who served as the U.S . Representative for Floridas 13th congressional district , based in Pinellas County , from 2014 to 2017 . He was subsequently reelected in November 2014 , winning 75 percent of the vote , but was unseated in 2016 by former Governor Charlie Crist after court-ordered redistricting made his district more Democratic . In September 2018 , Jolly announced he had left the Republican Party .
Extract information relevant to the query:
There is nothing relevant to the query.
Context: Early life . Jolly was born in Dunedin , Florida , the son of Judith and Lawson Jolly , a Baptist pastor . He received his B.A . degree from Emory University in 1994 and his J.D . degree from the George Mason University School of Law in 2001 .
Extract information relevant to the query:
```python
information.append({"subject": "David Jolly", "relation": "go to school", "object":
"Emory University", "time": {"start": datetime(1990, 1, 1), "end": datetime(1994, 12,
31)}})
information.append({"subject": "David Jolly", "relation": "go to school", "object":
"George Mason University", "time": {"start": datetime(1995, 1, 1), "end": datetime(2001,
12, 31)}})
```

Figure 4: TimeQA prompt part 1.

**Question: Which position did Crispin Blunt hold in Feb 2011?**

Question parsing:

```python
query = {"subject": "Crispin Blunt", "relation": "hold position", "object": None,
"time": {"start": datetime(2011, 2, 1), "end": datetime(2011, 2, 28)}}
answer_key = "object"
```

Search:

```python
entities_to_search = ["Crispin Blunt"]
```

Generate a background document from Wikipedia to answer the given question: Crispin Blunt is a British Conservative Party politician who has been the Member of Parliament (MP) for Reigate since 1997. In February 2011, he was appointed as the Parliamentary Under-Secretary of State for Prisons and Youth Justice in the Ministry of Justice.

Extract information relevant to the query:

```python
information.append({"subject": "Crispin Blunt", "relation": "hold position",
"object": "Member of Parliament", "time": {"start": datetime(1997, 1, 1), "end":
None}})
information.append({"subject": "Crispin Blunt", "relation": "hold position",
"object": "Parliamentary Under-Secretary of State for Prisons and Youth Justice",
"time": {"start": datetime(2011, 2, 1), "end": None}})
```

Context: Crispin Blunt Crispin Jeremy Rupert Blunt ( born 15 July 1960 ) is a British Conservative Party politician . He has served as Member of Parliament ( MP ) for Reigate since 1997 , and from May 2010 to September 2012 he was the Parliamentary Under-Secretary of State for Prisons and Youth Justice within the Ministry of Justice . Blunt first entered the House of Commons at the 1997 general election , when he replaced the then MP Sir George Gardiner who had been deselected by the Constituency Conservative Association Executive Council and joined the Referendum Party .

Extract information relevant to the query:

```python
information.append({"subject": "Crispin Blunt", "relation": "hold position",
"object": "Member of Parliament", "time": {"start": datetime(1997, 1, 1), "end":
None}})
information.append({"subject": "Crispin Blunt", "relation": "hold position",
"object": "Parliamentary Under-Secretary of State for Prisons and Youth Justice",
"time": {"start": datetime(2010, 5, 1), "end": datetime(2012, 9, 30)}})
```

Figure 5: TimeQA prompt part 2.

**Question:What's the capital of Klamath County, California between Aug 1854 and Jun 1855?**

Question parsing:
```python
query = {"subject": None, "relation": "capital of", "object": "Klamath County,
California", "time": {"start": datetime(1854, 8, 1), "end": datetime(1855, 6, 30)}}
answer_key = "subject"
```

Search:
```python
entities_to_search = ["Klamath County"]
```

Failed entities:
```python
failed_entities = ["Klamath County"]
similar_entities = ["Klamath County, California", "Klamath County, Oregon", "Klamath
Falls, Oregon"]
```

Search:
```python
entities_to_search = ["Klamath County, California"]
```

Generate a background document from Wikipedia to answer the given question: Klamath County, California was a county of California from 1851 to 1874. The county seat was Crescent City from 1851 to 1854 and then Yreka from 1854 to 1874.
Extract information relevant to the query:
```python
information.append({"subject": "Crescent City", "relation": "capital of", "object":
"Klamath County, California", "time": {"start": datetime(1851, 1, 1), "end":
datetime(1854, 12, 31)}})
information.append({"subject": "Yreka", "relation": "capital of", "object": "Klamath
County, California", "time": {"start": datetime(1854, 1, 1), "end": datetime(1874, 12,
31)}})
```

Context: Klamath County , California Klamath County was a county of California from 1851 to 1874 . During its existence , the county seat moved twice and ultimately portions of the territory it once had were carved up and added to nearby counties . The original county seat was Trinidad , on the countys southwestern coast . In 1854 the county seat was moved to Crescent City , because of its larger population . But the western portion of the county was unrepresentative of the mining interests in the eastern portion of the county , and so , in 1856 , the county seat was moved inland , to Orleans Bar , now Orleans . In 1857 , Del Norte County , including Crescent City , was split off from Klamath County . In 1874 Klamath County was finally abolished , divided between Siskiyou and Humboldt counties .
Extract information relevant to the query:
```python
information.append({"subject": "Trinidad", "relation": "capital of", "object": "Klamath
County, California", "time": {"start": datetime(1851, 1, 1), "end": datetime(1853, 12,
31)}})
information.append({"subject": "Crescent City", "relation": "capital of", "object":
"Klamath County, California", "time": {"start": datetime(1854, 1, 1), "end":
datetime(1855, 12, 31)}})
information.append({"subject": "Orleans", "relation": "capital of", "object": "Klamath
County, California", "time": {"start": datetime(1856, 1, 1), "end": datetime(1856, 12,
31)}})
```

Figure 6: TimeQA prompt part 3.