# OpenReview forum: "Question Answering as Programming for Solving Time-Sensitive Questions"
_EMNLP/2023/Conference — EMNLP 2023 Main_

### Official Review · Reviewer_LQzX · 2023-07-29

**Typos Grammar Style And Presentation Improvements:** The section headers casing is inconsi…
**Soundness:** 3

**Excitement:**

3: Ambivalent: It has merits (e.g., it reports state-of-the-art results, the idea is nice), but there are key weaknesses (e.g., it describes incremental work), and it can significantly benefit from another round of revision. However, I won't object to accepting it if my co-reviewers champion it.

**Paper Topic And Main Contributions:**

This paper introduces a new approach for answering questions with time constraints using LLMs. They propose to reformulate QA as a programming task. The LLM is prompted to generate a python dict. corresponding to the question and also extract some information in a structured form. Further, they use 'Check' and 'Match' functions to choose the correct answer. Results on standard datasets demonstrate the utility of this approach. The ablation studies show the usefulness of different components.

**Questions For The Authors:**

1. Could you elaborate on not using implicit questions in TempQuestions and TimeQuestions datasets for experiments?
2. How is the check function implemented? L461 mentions manual construction but no further description.
3. The wording in sec 3.3 is confusing. In L251-253, the authors say that they check whether items extracted from internal knowledge have shown up in external knowledge. The external and internal knowledge sources are different. What action is taken after making this check and why is it necessary?

**Reasons To Accept:**

1. The authors show consistent improvement on three datasets.
2. It shows an important application of LLMs augmented with tools for temporal reasoning.


**Reasons To Reject:**

1. The use of the coding ability of LLMs has been demonstrated in prior work [Gao et al. 2022b, Chen et al. 2022, Lyu et al. 2023] for other reasoning tasks, so the novelty is limited
2. The authors don't test their approach on implicit questions in TempQuestions and TimeQuestions datasets.

**Reproducibility:**

3: Could reproduce the results with some difficulty. The settings of parameters are underspecified or subjectively determined; the training/evaluation data are not widely available.

**Reviewer Confidence:**

4: Quite sure. I tried to check the important points carefully. It's unlikely, though conceivable, that I missed something that should affect my ratings.

---

> ### Author Rebuttal · Authors · 2023-08-28
>
> Thank you for your detailed review, insightful and constructive comments!
>
> ### On the novelty compared to prior works
>
> In response to Reason 1, we beg to differ, there are several differences between our work and the prior code-prompt works.
>
> i) Prior code-prompt works mainly utilized the coding ability of LLMs for solving mathematical and symbolic reasoning tasks, while we focus on solving time-sensitive factual questions with this coding ability, therefore the task is quite different.
>
> ii) Prior code-prompt works only used LLMs’ own knowledge for problem solving, while we propose to apply LLMs’ coding ability to represent both LLMs’ internal knowledge and external knowledge (e.g., Wikipedia articles) as same-format codes, this enables us to easily enhance LLMs with different sources of knowledge, and also facilitates desired processing.
>
> iii) Prior code-prompt works did not explore verifying the correctness of LLMs-generated codes, while we propose to incorporate Check and Match steps to mitigate LLMs’ hallucination and ensure accuracy, our experiments in Table 3 validate the importance of it. Our method requires very little human effort, and we also explore using LLMs for constructing those functions in Sec 4.3.4.
>
> We will add more clarification of difference between prior works.
>
> ### On not using implicit questions
>
> In response to Reason 2 and Question 1, implicit questions require extra steps to uncover the implicit time indicated by the things/events, while we only used a simple Wikipedia search engine and there are many ambiguous things that have the same name (e.g., song, movie, person name, …), it is difficult for LLMs to determine which entry corresponds to the thing mentioned in the question. Therefore, stronger retrievers are needed to retrieve relevant documents. It is worth noting that our method is agnostic to the way of retrieving external documents and can be seamlessly combined with those off-the-shelf retrievers as mentioned in L333-337. We’ll add these clarification in the revision, thanks for the pointer!
>
> ### Implementation of Check function
>
> In response to Question 2, we described what check in L247-254. Specifically, the logic of the Check function looks like the following pseudo code, we will open source all the codes in the next version.
>
> ```python
> import datetime
> query = ...
> answer_key = "object"
> ...
> documents_generated_by_llms = [context1]
> documents_from_wiki = [context2, context3, ...]
> information_from_llms = [item1]
> information_from_wiki = [item2, item3, ...]
>
> # Ensure the extracted items' time appear in the corresponding context, otherwise discard the item
> for item, context in zip(information_from_llms, documents_generated_by_llms):
> 	CHECK item["time"]["start"].year in context or item["time"]["end"].year in context
> for item, context in zip(information_from_wiki, documents_got_from_wiki):
> 	CHECK item["time"]["start"].year in context or item["time"]["end"].year in context
>
> # Ensure the items extracted from llms generated document appear in those extracted from wikipedia document, otherwise discard the item
> for item in information_from_llms:
> 	CHECK item[answer_key] in [x[answer_key] for x in information_from_wiki]
> ```
>
> ### Check internal and external knowledge
>
> In response to Question 3, internal knowledge refers to documents generated by LLMs, external knowledge refers to documents from Wikipedia, we discard the items extracted from internal knowledge if they do not appear in those extracted from external knowledge, for more details please refer to the code above to see how we implement it. This action is necessary because LLMs often generate erroneous facts, which is investigated in many prior works, one such example is also shown in Figure 3. Check step can mitigate LLMs hallucination and improve the EM score by up to 15.4 as demonstrated in Sec 4.3.2 and Table 3.
>
> We will add more clarification of difference between prior works and dataset choosing, add more details about how we implement the Check and Match functions, open source our codes and fix typos and in the next version, thanks for the suggestion!

---

### Official Review · Reviewer_NXcU · 2023-08-05

**Soundness:** 3

**Excitement:**

4: Strong: This paper deepens the understanding of some phenomenon or lowers the barriers to an existing research direction.

**Paper Topic And Main Contributions:**

The paper proposes a way of prompting LLMs to answer questions by first parsing questions and external knowledge into code (i.e. programming language) prior to prompting the LLM to match the parsed question to the external evidence/knowledge in order to answer the question. The authors compare the proposed method to two state-of-the-art prompting approaches—ReAct and CoT—on three datasets containing time-sensitive questions and beat both baselines. The paper is quite well-written, and easy to follow and the proposed approach shows a good improvement over ReAct and CoT. However, it is unclear how the findings generalize to other LLMs; it is possible that with more powerful LLMs, the improvement from the extra parsing step won't be as useful or that the approach does not generalize as well to other types of constraints (beside time constraints).

**Questions For The Authors:**

- Did you try parsing the question into any other structured form other than code to see how it performs compared to the proposed approach?

**Reasons To Accept:**

- The authors do a thorough analysis of the impact of the proposed approach on one kind of QA datasets (time-sensitive).
- The approach is tested on multiple (3) datasets with time-constraints and outperforms state-of-the-art (ReAct and CoT).

**Reasons To Reject:**

- It is not clear how the proposed method will generalize to other QA tasks with other constraints (or no constraints).
- It would be helpful to study how the presented approach would perform on different LLMs.

**Reproducibility:**

3: Could reproduce the results with some difficulty. The settings of parameters are underspecified or subjectively determined; the training/evaluation data are not widely available.

**Reviewer Confidence:**

4: Quite sure. I tried to check the important points carefully. It's unlikely, though conceivable, that I missed something that should affect my ratings.

---

> ### Author Rebuttal · Authors · 2023-08-28
>
> Thank you for your detailed review, insightful and constructive comments!
>
> ### On method may not be useful with more powerful LLMs
>
> Regarding ***“it is possible that with more powerful LLMs, the improvement from the extra parsing step won't be as useful”***, the model we use is gpt-3.5-turbo, which is probably the most powerful LLM except for GPT-4, but our experiments show that it still performs poorly on time-sensitive QA task. GPT-4, the most powerful LLM, also makes simple mistakes as shown in a recent work [1], indicating there is still a long way to go before we can rely on LLMs alone to solve any QA task. We believe that the performance of our method can scale with the power of the backbone model, and this is verified by our additional experimental results below using text-davinci-003.
>
> ### Generalization to other QA tasks
>
> In response to Reason 1, we focus on solving questions with time constraints, however, there is few QA dataset that is specifically constructed to incorporate strict constraints in the questions. Nevertheless, our method can be easily extended to cope with other types of constraints, we just need to represent the constraint into an appropriate class in python (we use datetime.datetime() to represent time), and define the metric that measures how well is the constraint satisfied in the Match function. When there is no constraint, Check and Match steps are not necessary and our method degenerates into a method similar to CoT and ReAct. We’ll add these details in the revision, thanks for the pointer!
>
> ### Performance on different LLMs
>
> | Method                   | TimeQA-Easy (EM/F1) | TimeQA-Hard (EM/F1) | TempQuestions (EM/F1) | TimeQuestions (EM/F1) |
> | :----------------------- | :------------------ | :------------------ | :-------------------- | :-------------------- |
> | CoT (text-davinci-003)   | 17.0/26.2           | 15.0/28.0           | 39.0/49.6             | 33.0/44.8             |
> | ReAct (text-davinci-003) | 30.0/42.7           | 26.0/37.4           | 33.0/42.2             | 22.0/31.6             |
> | QAaP (text-davinci-003)  | **38.0/50.1**           | **38.0/49.4**           | **47.5/52.5**             | **40.0/48.9**             |
>
> In response to Reason 2, due to the high cost of OpenAI api, we only reported results of gpt-3.5-turbo in our previous version. To address the concern raised by Reviewer NXcU, here we did small-scale experiments with text-davinci-003 (random seed set to 0, sampling 100 questions each dataset) and the results confirm the effectiveness of our method. We’ll add the additional results as above in the paper.
>
> ### Other types of structured form
>
> In response to Question1, we did not parse the question into other structured form because LLMs are pre-trained on texts and codes, it is natural to apply them to transform the inputs from natural language to programming language. If the question/relevant information is parsed into other structured form other than code, it may be more complex and require more human efforts to verify its truthfulness and alleviate hallucination, which plays a key role in the final performance as shown in Table 4. We leave applying other types of structured form for future work.
>
> [1] **How Language Model Hallucinations Can Snowball**

---

### Official Review · Reviewer_PAqi · 2023-08-06

**Soundness:** 3

**Excitement:**

4: Strong: This paper deepens the understanding of some phenomenon or lowers the barriers to an existing research direction.

**Paper Topic And Main Contributions:**

This paper explores the inability of LLMs to handle rigorous reasoning task like temporal question answering. The key idea is to reframe the QA task as programming task. So the model doesn’t need get the answers directly using the input. Instead, this paper reformulates the QA as programming. In details, authors parse the questions that is same to translate the question to be a query. Meanwhile, this paper prompts LLMs to extract the information based on the query. Finally, the matching step will choose the best answer using a match score. This is similar to ranking problem. In general, the whole idea is simple and clear. The key is to reformulate the question as a well-defined query.

**Questions For The Authors:**

Please check the weaknesses and provide your answers.

**Reasons To Accept:**

Strengths:

1.	This paper proposes a new way to solve the temporal question answering task. The key of new method is to reframe the QA task as the programming task.

2.	This paper evaluates the QAaP framework on multiple time-sensitive QA datasets and achieves significant improvement. And It is great to see many discussions in the paper.

**Reasons To Reject:**

Weaknesses:

1.	In the introduction, authors mentioned “To tackle this aforementioned challenges, … ability in natural language and programming language …”. But question is also a natural language.  So why do we need to reframe the QA to programming. The description of the challenge is not clear.

2.	I am still not so sure if it is suitable to call it “reframe QA task as programming”. It is a step of  “question to query” and another step of information extraction based on LLMs. It is more related to reformulate text question to a structured query and then do extraction. There are many papers that use the similar methods. It is necessary to do the comparisons with them here on both the difference explanation and experimental comparison. The provided results in the experiments section are not enough to prove your contribution.

**Reproducibility:**

3: Could reproduce the results with some difficulty. The settings of parameters are underspecified or subjectively determined; the training/evaluation data are not widely available.

**Reviewer Confidence:**

4: Quite sure. I tried to check the important points carefully. It's unlikely, though conceivable, that I missed something that should affect my ratings.

---

> ### Author Rebuttal · Authors · 2023-08-28
>
> Thank you for your detailed review, insightful and constructive comments!
>
> ### Description of the challenges and necessities
>
> In response to Question 1, regarding the **challenges** (denote by **C**) and the **necessities** (denote by **N**) of reframing the QA as programming:
>
> - **C1.** As mentioned in L062-065, LLMs struggle to comprehend the sequential, overlapping and inclusive relationships between dates, which can be attributed to their nature as probabilistic models and their inability to perform rigorous reasoning as symbolic systems.
> - **N1.** Though LLMs cannot rigorously reason by themselves, they are able to generate high-quality and executable codes, we endeavor to harness their strong coding ability to transform the question and context into well-structured codes, this avoids rigorous reasoning based on surface-level text semantics and facilitates the subsequent processing and reasoning over the collected code-format information (Check and Match steps).
> - **C2.** As mentioned in L066-073, it is difficult for LLMs to directly locate the relevant information and provide the correct answer, as relevant facts may often be scattered across a long document and expressed in diverse ways, this challenge is also proved by a recent work [1].
> - **N2.** In QAaP, we apply LLMs to transform document segments into same-format python dicts and save them in a python list, this allows a comprehensive gathering of the relevant details dispersed throughout (arbitrary number of) documents.
> - **C3.** As mentioned in L100-105, the answer directly provided by LLMs in natural language is hard to verify and is often erroneous, it is very likely that the answer does not satisfy the constraint and thus incorrect as shown in Table 2, Table 4 and Figure 3.
> - **N3.** Since all the obtained information is represented as codes, we can easily construct two functions Check and Match to reduce hallucination and ensure accuracy as mentioned in L091-099. These two steps are built on the previous steps and play a key role in the final performance as illustrated in Table 3 and 4, without which performance will drop 10+%.
>
> We will make modifications to make the description of challenges and necessities more clear and coherent in the next version, thanks for the pointer!
>
> ### Difference explanation and experimental comparison.
>
> In response to Question 2,
>
> i) Regarding ***“not so sure if it is suitable to call it “reframe QA task as programming””***, in QAaP we parse the question into a python dict, extract relevant information from documents and represent them as python dicts, the entire problem solving process is based on those codes and the answer is determined by code execution, therefore we call it “reframe QA task as programming”.
>
> ii) Regarding ***difference between prior works***, the most similar works are KBQA and MRC methods, there are several differences: 1) our method is few-shot, while they are mainly supervised; 2) we only use a simple Wikipedia search engine, while KBQA require high-quality annotated knowledge bases; 3) we are able to verify if the constraint is satisfied, while MRC methods cannot.
>
> iii) Regarding ***experimental comparison to prior works***, in Table 1 the supervised baseline TEQUILA and EXAQT for TempQuestions and TimeQuestions  are KBQA methods, the baseline for TimeQA is a MRC model. All of them are strong baselines that perform answer extraction based on the question and we have recognized that there is still a gap between few-shot methods and supervised methods in L373-376.
>
> We add clarification of differences between prior works and experimental comparison in the next version, thanks for the suggestion!
>
> [1] **Lost in the Middle: How Language Models Use Long Contexts**

---

### Meta-Review · Area_Chair_X494 · 2023-09-16

**Recommendation:** 5

**Metareview:**

**Summary:** The paper proposes a new paradigm in question answering (specific to questions requiring temporal reasoning) by reformulating the QA task as a program and then utilizing the enhanced code-reasoning and understanding capabilities of LLM to solve for the solution. The question and external knowledge are first parsed into a code query, and then the LLM is prompted to match the parsed question to the external evidence/knowledge in order to answer the question. Empirical evaluation on 3 QA datasets with time-sensitive questions shows that the proposed approach beats SOTA prompting baselines (ReAct and CoT) by significant margins.

Overall, all reviewers acknowledged the clear writing of the paper, and the novelty and simplicity of the proposed solution. The claims in the paper are supported well empirically with a solid evaluation performed over 3 datasets and compared to two strong baselines. Most of the concerns raised in the initial reviews were sufficiently addressed by the author response.

**Recommendations for Improvement:** My suggestion would be to include the additional experiments with other LLMs (text-davinci-003, and possibly more) and elaborating the reason for excluding implicit questions (of TempQuestions and TimeQuestions datasets) in the paper to strengthen it further.

---

### Decision · Program_Chairs · 2023-10-07

**Decision:**

Accept-Main

**Comment:**

**Summary:** The paper proposes a new paradigm in question answering (specific to questions requiring temporal reasoning) by reformulating the QA task as a program and then utilizing the enhanced code-reasoning and understanding capabilities of LLM to solve for the solution. The question and external knowledge are first parsed into a code query, and then the LLM is prompted to match the parsed question to the external evidence/knowledge in order to answer the question. Empirical evaluation on 3 QA datasets with time-sensitive questions shows that the proposed approach beats SOTA prompting baselines (ReAct and CoT) by significant margins.

Overall, all reviewers acknowledged the clear writing of the paper, and the novelty and simplicity of the proposed solution. The claims in the paper are supported well empirically with a solid evaluation performed over 3 datasets and compared to two strong baselines. Most of the concerns raised in the initial reviews were sufficiently addressed by the author response.

**Recommendations for Improvement:** My suggestion would be to include the additional experiments with other LLMs (text-davinci-003, and possibly more) and elaborating the reason for excluding implicit questions (of TempQuestions and TimeQuestions datasets) in the paper to strengthen it further.